# Comparison of Long-Term Effects between Chest Compression-Only CPR Training and Conventional CPR Training on CPR Skills among Police Officers

**DOI:** 10.3390/healthcare9010034

**Published:** 2021-01-02

**Authors:** Byung-Jun Cho, Seon-Rye Kim

**Affiliations:** 1Department of Emergency Medical Technology, College of Health Science, Kangwon National University, 346 Hwangjo-gil, Dogye-up, Samcheok-si, Gangwon-do, Samcheok 25945, Korea; cho6451@gmail.com; 2Department of Pharmacy, College of Pharmacy, Kangwon National University, Chuncheon Campus 1, Gangwondaehakgil, Chuncheon-si, Gangwon-do, Chuncheon 24341, Korea

**Keywords:** chest compression-only CPR, accuracy of CPR skills, police officers, retention of CPR skills

## Abstract

Despite of the changes of out-of-hospital cardiac arrest (OHCA) survival rise when bystander CPR is provided, this was only conducted in about 23% of OHCA patients in Korea in 2018. Police officers acting as first responders have a high chance of witnessing situations requiring CPR. We investigated long-term effects on CPR quality between chest compression-only CPR training and conventional CPR training in police officers to find an efficient CPR training method in a prospective, randomized, controlled trial. Police officers underwent randomization and received different CPR training. With the Brayden Pro application, we compared the accuracy of CPR skills immediately after training and the one after 3 months. Right after training, the conventional CPR group presented the accuracy of the CPR skills (compression rate: 74.6%, compression depth: 66.0%, recoil: 78.0%, compression position: 96.1%) and chest compression-only CPR group presented the accuracy of the CPR skills (compression rate: 74.5%, compression depth: 71.6%, recoil: 79.2%, compression position: 99.0%). Overall, both groups showed the good quality of CPR skills and had no meaningful difference right after the training. However, three months after training, overall accuracy of CPR skills decreased, a significant difference between two groups was observed for compression position (conventional CPR: 80.0%, chest compression only CPR: 95.0%). In multiple linear regression analysis, three months after CPR training, chest compression-only CPR training made CPR skills accuracy 28.5% higher. In conclusion, police officers showed good-quality CPR right after CPR training in both groups. But three months later, chest compression-only CPR training group had better retention of CPR skills. Therefore, chest compression-only CPR training is better to be a standard training method for police officers as first responders.

## 1. Introduction

As the society changes to an aged society, the number of patients with cardiac arrest, the major cause of death in Korea, is increasing. The mortality of out-of-hospital cardiac arrest(OHCA) remains high. A total of 29,832 OHCA victims per million people in South Korea are recorded in 2017 [1]. Since irreversible brain damage would start after four to five minutes of cardiac arrest, the survival rate of OHCA is inversely proportional to the time elapsed in starting cardiopulmonary resuscitation(CPR) after OHCA [2]. In Korea, 23.5% of OHCA victims received CPR from a bystander in 2018. It is lower than 33.3% in the United States [3]. CPR from the bystander is a key function in the chain of survival [4]. The survival rate of OHCA was about four times higher when the patients received CPR from bystanders, so increasing the rate of CPR from bystanders is especially important [5]. Thus, a strategy to increase the survival rates should focus on prompt CPR performance by bystanders [6]. 

Also, the quality of chest compressions performed is especially important. 2015 American Heart Association (AHA) guidelines recommend the appropriate compression rate and depth during performing CPR [7]. Lately, chest compression-only CPR has been urged as an alternative to conventional CPR, since chest compression-only CPR is simpler to learn and easier to perform than the conventional one [8,9,10,11,12,13,14]. There are some researches that people who got chest compression-only CPR training conducted high quality of chest compressions than those who got conventional CPR training [15,16,17,18,19]. The quality of chest compressions such as continuous chest compressions with proper depth is a key point of successful CPR [20,21]. 

Emergency Medical Service Act of Korea demonstrates that the police is regarded as first responders at the scene. Actually there was a report that the police arrived at the scene before 119 (the telephone number of the firefighter call center of Korea) [22]. So, it is needed to enhance the quality of CPR performance of police officers. The Act on Emergency Medical Services mentions about the establishment of standards in relation to CPR training for efficient performance, and the importance of CPR training can be confirmed by requiring the completion of retraining every year [23]. Previous studies reported acquired skills decreased in a period of 3 to 6 months, so the proper period should be considered for effective CPR retraining policy [24,25,26,27,28,29]. AHA also recommends retraining within two years after completing the first training in order to maintain the educational effect of CPR [4]. There are no consensus about how long the training effect maintains.

Generally CPR training for the public focuses on basic life support(BLS) knowledge and the quality of CPR skills. So, this study aims to compare long-term effects on CPR performance quality between chest compression-only CPR training and conventional CPR training among police officers, as not medical personnel but the first responders, and to emphasize the importance of CPR training by finding efficient CPR training methods. 

## 2. Materials and Methods

### 2.1. Study Design and Subjects

The study was designed as a randomized controlled trial. Volunteers were randomly assigned to either chest compression-only CPR training group or conventional CPR training group with stratification by sex and age using permuted blocks. A total of 119 police officers in Korea participated in the study. The inclusion criteria were to be police officers, having below ten times previous CPR training during the past 10 years, have availability to attend CPR training. Subjects who could not attend CPR training or the follow-up were excluded. Total of 59 police officers received 150 min training program which consisted of theoretical education (30 min), chest compressions practice (90 min), and automated external defibrillator (AED) operation (30 min). About 60 police officers received 180 min training program which consisted of theoretical education (30 min), chest compressions practice (90 min), pocket mask ventilations practice (30 min), and AED operation (30 min) on the basis of 2015 AHA guidelines. Both chest compression-only CPR training and conventional CPR training were carried out, and accuracy of their CPR skills was assessed 10 min (right) after CPR training without evaluating ventilations. Also the follow-up measurement without evaluating ventilations at 3 months after CPR training was conducted. All progressive steps were performed by following the Declaration of Helsinki. Volunteers submitted informed consent before joining in the study. The study was approved by the ethics committees of Nambu University (reference number 1041478-2017-BR-001). 

### 2.2. Data Collection

After the two kinds of CPR training, we evaluated the participants’ CPR skills. In this assessment, a manikin used to measure CPR skills’ accuracy was the Brayden Pro (Indonesian Inc., Seoul, Korea), which was set by the 2015 CPR guidelines of AHA. The Brayden Pro application on tablet PC was connected to the manikin then conducted evaluation mode. The data of CPR skills including chest compression rate, chest compression depth, chest compression rate accuracy, chest compression depth accuracy, recoil accuracy, and chest compression position accuracy were collected for 2 min and used as data for this study.

### 2.3. Measure Accuracy of CPR Skills 

The CPR training lasted for 150 min, which consisted of 30 min in CPR knowledge and 120 min in CPR practice. The CPR knowledge included contents related to BLS such as survival chain, the BLS procedure, and CPR, emphasizing the importance of high-quality CPR following guideline of AHA-2015 [2]. The Brayden Pro application checked average rates and depths of chest compressions for 2 min during CPR performance, and measured accuracy of every compression. The accuracy of CPR skills defined as proportion of chest compressions with accurate rate, accurate depth, accurate recoil and accurate position among the total chest compressions. The accurate depth was defined as 5.0–6.0 cm, the accurate rate was defined as 100–120 compressions per minute, the accurate recoil was defined as 100% recovery, and the accurate position indicated below a half point of sternum according to the 2015 AHA CPR guideline [2]. 

### 2.4. Other Variables

In this study, we tested CPR knowledge to perform CPR. It was composed of 8 questions: calling 119 (emergency phone) for help, consciousness check, breath check, chest compression position, airway maintenance, correct proportion of compressions-ventilation, depth and frequency of chest compressions, and irreversible brain damage starting time. Each question was scored 1 point for a correct answer and 0 point for a wrong answer. We asked the participants for self-reporting confidence to CPR performance in real situations. It was composed of four questions: Can you perform CPR when you find CA victims? Can you perform CPR for your family? Are you confident for performing CPR? Can you use AED for CA victims? The answer to each question was evaluated as 1 point for yes and 0 point for no, and the higher the score from 0 to 4 points, the greater the confidence in CPR performance. Also we checked participants’ attitude to performing CPR in real situations, and this variable was measured with 5 point Likert scale. 

### 2.5. Statistical Analysis

The distribution was normal as a result of the analysis, and the variables were presented as mean (standard deviation) or frequencies(%). T-tests were used to compare the variables such as knowledge, confidence, attitude that determine CPR quality variables between two groups right after training and 3 months later. The multiple linear regression analysis was carried out to measure the impact factors on accuracy of CPR skills right after training and 3 months later. We used SPSS for Windows ver. 20.0 (IBM Co., Armonk, NY, USA), and all analysis were performed with the significance level of *p* < 0.05. 

## 3. Results

### 3.1. Homogeneity in Demographic Characteristics

According to gender, 50 (83.3%) participants of the conventional CPR group were male, 50 (84.7%) participants of the chest compression-only CPR group were male. The mean age of conventional CPR group was 41.7 ± 11.0 years, and that of chest compression-only CPR group was 42.9 ± 12.3 years. The mean number of CPR training during the past ten years in the conventional CPR group was 4.3 ± 4.5, and that of chest compression-only CPR group was 4.2 ± 4.7. There were 36 (60.0%) persons, whose elapsed duration was less than one year after the last CPR training in the conventional CPR group (Table 1). 

### 3.2. Knowledge, Confidence, Attitude, and CPR Skills at Right after CPR Training

Concerning the score of knowledge, the conventional CPR group was 5.9 ± 1.2, while the chest compression-only CPR was 6.0 ± 1.0. About confidence of CPR, the score of conventional CPR group was 3.3 ± 0.8 and that of chest compression-only CPR was 3.4 ± 0.9. About attitude to CPR, the score of conventional CPR group was 4.1 ± 0.6 and that of chest compression-only CPR was 4.0 ± 0.5. The compression rate of the conventional CPR group was 112.4 ± 5.7/min, and that of chest compression-only CPR was 116.1 ± 4.7/min. The compression depth of the conventional CPR group was 5.5 ± 0.4 cm, while that of chest compression-only CPR was 5.5 ± 0.5 cm. The accuracy of compression rate of the conventional CPR group was 74.6 ± 26.6%, and that of chest compression-only CPR group was 74.5 ± 24.4%. The accuracy of compression depth of the conventional CPR group was 66.0 ± 27.2%, and that of chest compression-only CPR was 71.6 ± 33.9%. The accuracy of recoil of conventional CPR group was 78.0 ± 27.1%, and that of chest compression-only CPR was 79.2 ± 29.6%. Lastly, the accuracy of compression position of the conventional CPR group was 96.1 ± 8.4%, and that of chest compression-only CPR was 99.0 ± 2.3% (Table 2). 

### 3.3. Knowledge, Confidence, Attitude, and CPR Skills at 3 Months after CPR Training

Concerning the score of knowledge, the score of the conventional CPR group was 5.5 ± 1.5, and the chest compression-only CPR group was 5.9 ± 1.2. About confidence of CPR, the score of the conventional CPR group was 2.9 ± 1.0, and the chest compression-only CPR group was 3.5 ± 0.6. About attitude to CPR, the score of the conventional CPR group was 3.6 ± 0.7, and the chest compression-only CPR group was 4.0 ± 0.5. The compression rate of the conventional CPR group was 123.1 ± 14.0/min, and that of the chest compression-only CPR group was 117.0 ± 6.4/min. The compression depth of the conventional CPR group was 5.9 ± 0.6 cm, and that of the chest compression-only CPR group was 5.8 ± 0.6 cm. The accuracy of compression rate of the conventional CPR group was 28.6 ± 26.6% and that of the chest compression-only CPR group was 35.1 ± 34.2%. The accuracy of compression depth of the conventional CPR group was 35.0 ± 32.2%, and that of the chest compression-only CPR group was 37.8 ± 36.3%. The accuracy of recoil of the conventional CPR group was 67.9 ± 35.2% and that of the chest compression-only CPR group was 69.1 ± 33.2%. The accuracy of compression position of the conventional CPR group was 80.0 ± 31.2%, and that of the chest-compression-only CPR was 95.0 ± 11.9%. There were significant differences in attitude to CPR, compression rate and accuracy of compression position (Table 3). 

### 3.4. Effects of Participants’ Characteristics on the Accuracy of CPR Skills Right after Training

Table 4 is the result of the multiple linear regressions of CPR skills accuracy right after CPR training. Multiple linear regression analysis built a significant Model I, F = 2.280, *p* = 0.031, explaining 16.3% of the variance in accuracy of CPR skills (R^2^ = 0.163). There were no significant impact factors. Gender, age, previous CPR training, knowledge, confidence of CPR, attitude to CPR, and CPR training methods did not affect the accuracy of CPR skills (Table 4).

### 3.5. Effects of Participants’ Characteristics on the Accuracy of CPR Skills at 3 Months after Training

Table 5 shows the results of multiple linear regressions of CPR skills accuracy at 3 months after CPR training. Multiple linear regression analysis built a meaningful Model II with F = 2.629, *p* = 0.017, explaining 18.1% of the variance in accuracy of CPR skills (R^2^ = 0.181). Age was negatively correlated with participants’ CPR skills, indicating that older age reduced CPR skills quality. Chest compression-only CPR training compared to conventional CPR training was positively correlated with participants’ CPR skills, enhancing accuracy of CPR skills. Chest compression-only CPR training increased the quality of CPR skills not right after training but at 3 months after training. Gender, previous CPR training, knowledge, confidence of CPR, and attitude to CPR were not correlated with participants’ CPR skills. Significant predictors for a higher CPR performance were younger age and chest compression-only CPR training (Table 5).

## 4. Discussion

Although the police officers are not professional medical personnel, they are highly likely to perform CPR on the spot as first responders [22]. The police officers take CPR training for the public, but CPR training program for the public has fewer training sessions and longer retraining interval, compared to training program for professional medical personnel. Therefore, in order to find efficient CPR training method for police officers as first responders, this study compared long-term effects on CPR performance quality between chest compression-only CPR training and conventional CPR training among police officers. In two ways, the immediate effect and the retention of training were verified by repeatedly measuring the accuracy of skills right after training and at 3 months after training. Also, the participants were asked to report their knowledge, confidence, and attitude to performing CPR right after and at 3 months after training. This study demonstrated that chest compression-only CPR training was more efficient than conventional CPR training.

### 4.1. CPR Skills after Training

Right after CPR training, all participants showed good CPR skills. The quality of CPR skills did not show any difference between chest compression-only CPR training group and conventional CPR training group right after CPR training. The high quality of CPR was due to the accuracy of compression rate and compression position. Most participants obtained good results, which were similar to previous studies. All subjects performed with correct compression position [30], depth of 50–60 mm [31,32], and rate of 100–120 compressions/min [33,34]. In addition, most of our participants reported a good knowledge, high confidence, and positive attitude, demonstrating a prompt positive effect of CPR training. There was no significant difference between two groups. Previous studies reported that the effect of recent CPR training on attitude was shown from 74.7% in pre-CPR to 78.8% in post-CPR [35]. But at 3 months after CPR training, all participants showed decreased CPR skills. Although participants showed lower quality of CPR skills at 3 months after CPR training, we found significant differences in compression rate and accuracy of compression position between the two groups. The quality of CPR skills in chest compression-only CPR training group was better than that in the conventional CPR training group at 3 months after CPR training. There was a major decrease in the accuracy of compression rate and compression depth at 3 months after CPR training, which resulted in low-quality CPR. As a previous study reported that this modality was likely to deepen at four months after BLS training [36], all trainees should be trained in every training session with emphasis on adequate compression rate and depth. Also, the score of attitude to CPR in chest compression-only CPR training group was higher than that in conventional CPR training group at 3 months after CPR training. Nevertheless, the most participants got good knowledge and high confidence of CPR without group’s difference. But other research reported different result that retaining knowledge on these BLS steps decreased in long term, like after 8 months [37,38]. Previous studies have showed that confidence increases significantly after CPR training [39,40], decrease to 70% after all [39]. Low confidence and competence might affect decision to perform CPR in emergency, which indicates the need for short and periodic training [41].

### 4.2. Multiple Linear Regression Analysis 

In multiple linear regression analysis, we found time-dependent difference in the accuracy of CPR skills. There was no significant impact variable on the accuracy of CPR skills right after CPR training. But impact variables on the accuracy of CPR skills at 3 months after CPR training were age and CPR training method. Younger age enhanced the CPR quality at 3 months after training. Older age decreased accuracy of CPR skills, as previous researches showed the result that younger participants had better retention [37,42,43,44]. Repeating training can help trainees to retain knowledge of CPR and to achieve high-quality performance of CPR [42]. In this study, the frequency of previous CPR training, knowledge of CPR, confidence and attitude to CPR had a positive effect. In a previous study, as CPR skills of professional nurses were assessed at regular intervals, retraining was found to improve knowledge and skills [43]. Similarly, lack of training was a main factor that negatively influenced the attitudes toward the performance of CPR in previous studies [42,45]. However, what is more important is retention of knowledge and skills of CPR requires systematic training methodology [37]. In our study, chest compression-only CPR training group had higher accuracy of CPR skills than conventional CPR training group. Therefore, regarding retention of CPR skills, chest compression-only CPR training could be more effective for non-professionals.

In addition, previous studies reported that the chest compression-only CPR training might be easier to learn and perform [14,21,46,47,48], so it could increase the number of bystander who would conduct bystander-initiated CPR [14,48,49,50,51]. According to a study by Stein et al., the Zurich Police Station made CPR and AED education mandatory for police officers in 2009. After placing trained police on the site, on-set time of CPR performance and using AED were significantly faster, and survival rates were significantly increased. In this way, detailed legal standards for CPR training contents including AED education, training hours, and training methods should be prepared to get actual results [47,48,49,50,51,52,53].

Our results indicated that CPR training was beneficial, and that there is a retention difference of CPR skills between conventional CPR training and chest compression-only CPR training. If focusing on retention of CPR skills, the CPR training method would be important. The effectiveness of CPR training might be enhanced by applying efficient training methods. So, chest compression-only CPR training should be applied for proper CPR performance. The police officers could frequently encounter cardiac arrest victims in the field. Therefore, we suggested that chest compression-only CPR training would be the standard training method for the public including police officers in order to grow bystander CPR and improve quality of CPR skills.

The study was not free from limitations. Limits contained subjective characteristics of the questionnaire responses and the possibility of sampling bias. Our participants were recruited from one province, therefore, there should be some bias related with characteristics such as height, weight, economic level, and educational level. However, all variables were adjusted for determining the impact factors on the accuracy of CPR skills by using the multiple linear regression analysis. Also, as we used manikins with simulated environment, this result could not be deduced to real situations. If the ventilations were assessed in the simulation, it is likely that the overall quality was higher in the conventional CPR training group. But the assessment of the chest compression-only was done. So there is the limitation of overall quality. Furthermore, this study was a 3-month longitudinal study, which affected the sample size.

## 5. Conclusions

Police officers as first responders should effectively perform CPR at the field for OHCA. Right after training, most police officers who took either chest compression-only CPR training or conventional CPR training performed appropriate CPR. However, at 3 months after training, the accuracy of CPR skills in chest compression-only CPR training group was higher than that in the conventional CPR training group. Furthermore impact factors on accuracy of CPR skills at 3 months after training were age and CPR training method. Regarding that results, we suggested that chest compression-only CPR training should be the standard CPR training method for police officers.

## Figures and Tables

**Table 1 healthcare-09-00034-t001:** Homogeneity demographic characteristics.

Characteristics		Conventional CPR (n = 60) n (%) or M ± SD	Chest Compression-Only CPR (n = 59) n (%) or M ± SD	*p*
Gender	Male	50 (83.3%)	50 (84.7%)	0.695
	Female	10 (16.7%)	9 (15.3%)	
Age(years)		41.7 ± 11.0	42.9 ± 12.3	0.780
Frequency of CPR training during the past ten years(n)		4.3 ± 4.5	4.2 ± 4.7	0.564
Elapsed duration from last CPR training	Under 1 year	36 (60.0%)	35 (59.3%)	0.312
	Above 1 year	24 (40.0%)	24 (40.7%)	

CPR (cardiopulmonary resuscitation), M (mean), SD (standard deviation).

**Table 2 healthcare-09-00034-t002:** Knowledge, confidence, attitude, and CPR skills right after CPR training.

Characteristics	Conventional CPR (n = 60) M ± SD	Chest Compression-Only CPR (n = 59) M ± SD	*p*
Knowledge of CPR(score)	5.9 ± 1.2	6.0 ± 1.0	0.559
Confidence of CPR(score)	3.3 ± 0.8	3.4 ± 0.9	0.651
Attitude to CPR(score)	4.1 ± 0.6	4.0 ± 0.5	0.314
Compression Rate (compressions/minute)	112.4 ± 5.7	116.1 ± 4.7	0.041 *
Compression Depth (cm)	5.5 ± 0.4	5.5 ± 0.5	0.826
CPR skills’ accuracy			
Accuracy of Compression Rate (%)	74.6 ± 26.6	74.5 ± 24.4	0.990
Accuracy of Compression Depth (%)	66.0 ± 27.2	71.6 ± 33.9	0.576
Accuracy of Recoil (%)	78.0 ± 27.1	79.2 ± 29.6	0.896
Accuracy of Compression Position (%)	96.1 ± 8.4	99.0 ± 2.3	0.162

CPR (cardiopulmonary resuscitation), M (mean), SD (standard deviation), * (*p* < 0.05).

**Table 3 healthcare-09-00034-t003:** Knowledge, confidence, attitude, and CPR skills at 3 months after CPR training.

Characteristics	Conventional CPR (n = 60) M ± SD	Chest Compression-Only CPR (n = 59) M ± SD	*p*
Knowledge of CPR(score)	5.5 ± 1.5	5.9 ± 1.2	0.369
Confidence of CPR(score)	2.9 ± 1.0	3.5 ± 0.6	0.055
Attitude to CPR(score)	3.6 ± 0.7	4.0 ± 0.5	0.014 *
Compression Rate (compressions/minute)	123.1 ± 14.0	117.0 ± 6.4	0.039 *
Compression Depth (cm)	5.9 ± 0.6	5.8 ± 0.6	0.891
CPR skills’ accuracy			
Accuracy of Compression Rate (%)	28.6 ± 26.6	35.1 ± 34.2	0.413
Accuracy of Compression Depth (%)	35.0 ± 32.2	37.8 ± 36.3	0.756
Accuracy of Recoil (%)	67.9 ± 35.2	69.1 ± 33.2	0.875
Accuracy of Compression Position (%)	80.0 ± 31.2	95.0 ± 11.9	0.020 *

CPR (cardiopulmonary resuscitation), M (mean), SD (standard deviation), * (*p* < 0.05).

**Table 4 healthcare-09-00034-t004:** The effects of participants’ characteristics on the accuracy of CPR skills at right after training.

Characteristics	Model I				
	B	S. E.	Beta	t	*p*
Constant	393.182	79.235		4.962	0.000
Gender (male vs. female)	1.220	16.721	0.009	0.073	0.942
Age(years)	−4.109	2.144	−0.246	−1.916	0.061
Previous CPR training(n)	1.178	16.656	0.009	0.071	0.944
Knowledge of CPR (score)	5.486	8.029	0.094	0.683	0.498
Confidence of CPR (score)	6.114	10.802	0.080	0.566	0.574
Attitude to CPR(score)	19.587	15.331	0.176	1.278	0.207
CPR Training Methods(compression-only vs. conventional)	0.545	8.165	0.009	0.067	0.947

CPR (cardiopulmonary resuscitation), S.E. (standard error). Model I: Adjusted R square = 0.163, F = 2.280, *p* = 0.031.

**Table 5 healthcare-09-00034-t005:** The effects of participants’ characteristics on the accuracy of CPR skills at 3 months after training.

Characteristics	Model II				
	B	S. E.	Beta	t	*p*
Constant	195.016	62.789		3.106	0.003
Gender (male vs. female)	−3.389	1.724	−0.241	−1.965	0.055
Age(years)	−1.072	0.450	−0.296	−2.381	0.021 *
Previous CPR training(n)	4.481	14.404	0.040	0.311	0.757
Knowledge of CPR (score)	2.963	6.464	0.061	0.311	0.649
Confidence of CPR (score)	6.110	8.797	0.094	0.458	0.491
Attitude to CPR(score)	9.732	12.609	0.106	0.695	0.444
CPR Training Methods(compression-only vs. conventional)	28.448	13.792	0.262	0.416	0.044 *

CPR (cardiopulmonary resuscitation), S.E. (standard error), * (*p* < 0.05). Model II: Adjusted R square = 0.181, F = 2.629, *p* = 0.017.

## Data Availability

No new data were created or analyzed in this study. Data sharing is not applicable to this article.

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
