# Peer review of "Comparison of Long-Term Effects between Chest Compression-Only CPR Training and Conventional CPR Training on CPR Skills among Police Officers"

_healthcare, 2021, doi:10.3390/healthcare9010034_

Round 1

Reviewer 1 Report

Dear authors.
You have done a good job of revision and modification, taking into account that, in the previous version, your manuscript was rejected.
I have reviewed all the aspects that I previously indicated to you that needed profound changes. Those changes are well fixed and I don't have much more to add.
The only recommendation that I make to you is that you can cite a reference on CPR Hands-only from the latest published in scientific journals, as it is convenient for discussion. I suggest to you that you can include the following reference in the discussion:
Maria José Pujalte-jesús, José Luis Díaz Agea, César Leal-costa. Is mouth-to-mouth ventilation effective in fi rst responders? Comparing the effects between 30: 2 algorithm versus hands-only. An exploratory pilot simulation study. Signa Vitae. 2020.doi: 10.22514 / sv.2020.16.0062.

Author Response

Dear authors.
You have done a good job of revision and modification, taking into account that, in the previous version, your manuscript was rejected.
I have reviewed all the aspects that I previously indicated to you that needed profound changes. Those changes are well fixed and I don't have much more to add.
The only recommendation that I make to you is that you can cite a reference on CPR Hands-only from the latest published in scientific journals, as it is convenient for discussion. I suggest to you that you can include the following reference in the discussion:
Maria José Pujalte-jesús, José Luis Díaz Agea, César Leal-costa. Is mouth-to-mouth ventilation effective in fi rst responders? Comparing the effects between 30: 2 algorithm versus hands-only. An exploratory pilot simulation study. Signa Vitae. 2020.doi: 10.22514 / sv.2020.16.0062.

==> Thank you for your concession.

I included above article in the discussion.

Reviewer 2 Report

Dear Authors,

Thank you for integrating some corrections of English grammar and style mistakes in your updated manuscript. However, I still believe it needs extensive English editing before a reviewer can make an opinion on whether the contents themselves hold merit for publication to a broader (international, English-speaking) audience.

I have "corrected" your Abstract as follows, but have not done so for the main text (this is not the purpose of a peer-review!).

  • Abstract and text: Always explain an abbreviation the first time you use it (also in the abstract)!
  • Abstract: Change the first sentence to "Despite of the changes of OHCA survival rise when bystander CPR is provided, this was only conducted in about 23% of OHCA patients in Korea in 2018."
  • Abstract: Change the second sentence to: "Police officers acting as first responders have a high chance of witnessing situations requiring CPR."
  • Line 16: Delete "119".
  • Line 18: Delete "At".
  • Line 19: Change to "THE conventional CPR group".
  • Line 19-22: This sentence is still not gramatically correct and does not make sense.
  • What is meant by "Both groups showed good quality and no difference" - This sentence is far too vague.
  • Line 23: Delete "At".
  • Line 24: Change to "[...] a significant difference [...]"
  • Abstract, general: I highly recommend structuring the Abstract in "Background", "Methods", "Results" and "Conclusion"!
  • The main text is still not suitably revised so that it is written in adequate English. For instance, the first sentence of the Introduction shows several mistakes. Apart from definitive mistakes, the language style still feels somewhat not right - As stated several times before, a native English speaker or a professional native English speaking revision service would solve this problem in very little time!

Author Response

Dear Authors,

Thank you for integrating some corrections of English grammar and style mistakes in your updated manuscript. However, I still believe it needs extensive English editing before a reviewer can make an opinion on whether the contents themselves hold merit for publication to a broader (international, English-speaking) audience.

I have "corrected" your Abstract as follows, but have not done so for the main text (this is not the purpose of a peer-review!).

  • Abstract and text: Always explain an abbreviation the first time you use it (also in the abstract)!
  • ==>I explained an abbreviation the first time.
  •  
  • Abstract: Change the first sentence to "Despite of the changes of OHCA survival rise when bystander CPR is provided, this was only conducted in about 23% of OHCA patients in Korea in 2018."
  • ==> I corrected the first sentence.
  •  
  • Abstract: Change the second sentence to: "Police officers acting as first responders have a high chance of witnessing situations requiring CPR."
  • ==> I corrected the second sentence.
  •  
  • Line 16: Delete "119".
  • ==> I deleted "119".
  •  
  • Line 18: Delete "At".
  • ==> I deleted "At".
  •  
  • Line 19: Change to "THE conventional CPR group".
  • ==> I changed to "The conventional CPR group".
  •  
  • Line 19-22: This sentence is still not gramatically correct and does not make sense. What is meant by "Both groups showed good quality and no difference" - This sentence is far too vague.
  • ==> I chaged to "Overall, both groups showed the good quality of CPR skills and had no meaningful difference right after the training."
  •  
  • Line 23: Delete "At".
  • ==> I deleted "At".
  •  
  • Line 24: Change to "[...] a significant difference [...]"
  • ==> I chaged to  "[...] a significant difference [...]"
  • Abstract, general: I highly recommend structuring the Abstract in "Background", "Methods", "Results" and "Conclusion"!
  • ==> I structured again the Abstract in "Background", "Methods", "Results" and "Conclusion".
  •  
  • The main text is still not suitably revised so that it is written in adequate English. For instance, the first sentence of the Introduction shows several mistakes. Apart from definitive mistakes, the language style still feels somewhat not right - As stated several times before, a native English speaker or a professional native English speaking revision service would solve this problem in very little time!
  • ==> A professional English speaker edited all text again.

This manuscript is a resubmission of an earlier submission. The following is a list of the peer review reports and author responses from that submission.

Round 1

Reviewer 1 Report

Dear authors.
The manuscript reports a very interesting research on CPR in first responders (police). Despite being an interesting and well-conducted study, I think it can be improved on the following aspects to be published in Healthcare.
- Line 57, it says 119, and I think it means 911.
- Reference 23 is divided into 2 in the bibliography (22 and 23).
- This study is very similar to this: https://pubmed.ncbi.nlm.nih.gov/29753857/ For some time, a maximum recycling time of 6 months has been established.
- I do not understand very well why "knowledge, confidence and attitude" have been measured, it seems to me that it does not provide information, and I do not know the validity of the questionnaire, or what was asked.
- You say that the groups have been randomized but you do not say how.
- Was the prior knowledge of the police officers in SVB assessed? Have they never received CPR training? In the results, you speak of "the average number of previous CPR trainings", but I don't know what it refers to or what it means, if they are regulated trainings, police courses on their own, or carried out by the police station.
- I have not seen inclusion / exclusion criteria: Table 1 says that 60% have received CPR training less than 1 year ago; Wouldn't it have been interesting to exclude those with recycling for less than 3 months? Was this taken into account?
- I think it would be interesting to have a comparative graph in which to see at first glance the differences in the two evaluation periods.
- There is a lot of information that is repeated in the results and tables.
- You value two methods of CPR (only compressions and the standard) but do not speak at any time about the ventilations. It would be interesting to know if this ability was maintained or decreased as much or more than compressions.
- In the results, the depth data (for example) do not vary much in cm, however, in% accuracy they vary a lot, I do not find much sense.

Best regards

Author Response

The manuscript reports a very interesting research on CPR in first responders (police). Despite being an interesting and well-conducted study, I think it can be improved on the following aspects to be published in Healthcare.
- Line 57, it says 119, and I think it means 911.

=>I added "119 (the telephone number of the firefighter call center of Korea) "

- Reference 23 is divided into 2 in the bibliography (22 and 23).

=>I divided into 2.

- This study is very similar to this: https://pubmed.ncbi.nlm.nih.gov/29753857/ For some time, a maximum recycling time of 6 months has been established.

=>I tried to verify various variables including different retraining period, different bystander group, different CPR training method, psychological variable such as confidence and attitude. So, my study is worth.

- I do not understand very well why "knowledge, confidence and attitude" have been measured, it seems to me that it does not provide information, and I do not know the validity of the questionnaire, or what was asked.

=>I added the questionnaire.

- You say that the groups have been randomized but you do not say how.

=> I used permuted blocks for randomization.

- Was the prior knowledge of the police officers in SVB assessed? Have they never received CPR training? In the results, you speak of "the average number of previous CPR trainings", but I don't know what it refers to or what it means, if they are regulated trainings, police courses on their own, or carried out by the police station.

=>In Korea, police officers are encouraged to take CPR training once a year. But it is not compulsory. Nowadadys regulated trainings were carried out by the police station.

- I have not seen inclusion / exclusion criteria: Table 1 says that 60% have received CPR training less than 1 year ago; Wouldn't it have been interesting to exclude those with recycling for less than 3 months? Was this taken into account?

=>I added inclusion / exclusion criteria.

- I think it would be interesting to have a comparative graph in which to see at first glance the differences in the two evaluation periods.

=> I agreed your opinion,

-There is a lot of information that is repeated in the results and tables.

=> I checked those and I edited.

-You value two methods of CPR (only compressions and the standard) but do not speak at any time about the ventilations. It would be interesting to know if this ability was maintained or decreased as much or more than compressions.

=> That's a good idea. But I didn't measure the ventilation. I will try to study next time.

-In the results, the depth data (for example) do not vary much in cm, however, in% accuracy they vary a lot, I do not find much sense.

=> Because the accuracy of depth was evaluated by every single compression, although mean of depth  do not vary much in cm, frequency of accurate the depth data could make big difference. And please consider the recommended rage, 5.0cm-6.0cm. If one depth was 4.9cm, the accuracy would be 0%. As calculation was performed Like this, the difference of n% might increase.

Reviewer 2 Report

I thank the authors for giving insight in their interesting study data on police officers being trained either in chest compressions only CPR or conventional CPR. I think that the data could be interesting for a broader audience, but I have substantial concerns about the manuscript in its current version.

-) Overall: The entire manuscript is in need of an English spelling and grammar review by a native speaker.

-) Title: I would amend the title to "[...] and conventional CPR training on CPR skills [...]" in order to make it easier to read.

-) Abstract: Try to shorten the abstract an structure it. As it is, it is too confusing for a potential reader and will not attract enough interest to read the full article. For instance, the first two sentences basically carry the same information.

-) Introduction: What is an "acute" CA patient? CA is always acute.

-) Introduction: When a person deceases, there will always be a CA - so the first sentence of the Introduction is confusing (CA as a main cause of death?).

-) Introduction: What exactly do you mean by "various stress situations"?

-) Introduction (line 38): Hypoxic brain damage begins right after CA, the question is only of how reversible it is.

-) Introduction (line 46): Chest compression quality is one of THE factors inducing outcome, it is wrong to say it "might" be especially important.

-) Introduction (lines 46ff): Please cite recent guidelines (e.g., 2015, or the new 2020 ILCOR recommendations).

-) Introduction (line 55): What is the Emergency Medical Service Act? You should not assume an international reader knows details about your system.

-) Introduction (lines 65ff): This paragraph appears too long and confusing, try to state the gaps of knowledge and your study aim in one or two sentences.

-) Methods (line 82): What kind of ventilation?

-) Methods: Many details about the inclusion and exclusion criteria are missing, e.g., had the participants had training before or not? I see that you included some information about this in the Results, but it should be stated here also.

-) Methods: When exactly took the measurement of CPR skills place (how long after training)? Please give details.

-) Methods (lines 110ff): Please give the exact questions. Also, what did you ask about brain damage as this seems a strange question for BLS training?

-) Results (line 128): Do not begin a sentence with a number.

-) Results (line 138): What is meant by "knowledge", "confidence of CPR" etc.? You do not define those terms beforehand, so you cannot expect a reader to follow your thoughts.

-) Discussion: Try to structure your Discussion more, maybe with subheadings. Try to group ideas in "packages" and try to follow a red line throughout the text so a reader can follow your thoughts easily without becoming confused or frustrated.

-) Discussion (lines 201-202): You are right, but statements like these must be cited.

-) Conclusion: Try to make a statement that one can understand without having read the whole article beforehand (e.g., "of both groups" - which groups? etc.).

Author Response

I thank the authors for giving insight in their interesting study data on police officers being trained either in chest compressions only CPR or conventional CPR. I think that the data could be interesting for a broader audience, but I have substantial concerns about the manuscript in its current version.

-) Overall: The entire manuscript is in need of an English spelling and grammar review by a native speaker.

=> The entire manuscript was reviewed by a native speaker.

-) Title: I would amend the title to "[...] and conventional CPR training on CPR skills [...]" in order to make it easier to read.

=>I changed "conventional CPR training on accuracy of CPR Skills " to "conventional CPR training on CPR skills". 

-) Abstract: Try to shorten the abstract an structure it. As it is, it is too confusing for a potential reader and will not attract enough interest to read the full article. For instance, the first two sentences basically carry the same information.

=> I deleted the first sentence.

-) Introduction: What is an "acute" CA patient? CA is always acute.

=>I deleted "acute".

-) Introduction: When a person deceases, there will always be a CA - so the first sentence of the Introduction is confusing (CA as a main cause of death?).

=>I changed like "as a main cause of death".

-) Introduction: What exactly do you mean by "various stress situations"?

=>I deleted "various stress situations".

-) Introduction (line 38): Hypoxic brain damage begins right after CA, the question is only of how reversible it is.

=>I added "irreversible".

-) Introduction (line 46): Chest compression quality is one of THE factors inducing outcome, it is wrong to say it "might" be especially important.

=>I changed "might be" to "is".

-) Introduction (lines 46ff): Please cite recent guidelines (e.g., 2015, or the new 2020 ILCOR recommendations).

=>I changed to 2015 guidelines.

-) Introduction (line 55): What is the Emergency Medical Service Act? You should not assume an international reader knows details about your system.

=>I added " of Korea".

-) Introduction (lines 65ff): This paragraph appears too long and confusing, try to state the gaps of knowledge and your study aim in one or two sentences.

=>I made short paragraph in two sentences.

-) Methods (line 82): What kind of ventilation?

=>pocket mask ventilations 

-) Methods: Many details about the inclusion and exclusion criteria are missing, e.g., had the participants had training before or not? I see that you included some information about this in the Results, but it should be stated here also.

=> I added details about the inclusion and exclusion criteria. 

-) Methods: When exactly took the measurement of CPR skills place (how long after training)? Please give details.

=> I added "10 minutes".

-) Methods (lines 110ff): Please give the exact questions. Also, what did you ask about brain damage as this seems a strange question for BLS training?

=>

-) Results (line 128): Do not begin a sentence with a number.

=>I added " According to gender"

-) Results (line 138): What is meant by "knowledge", "confidence of CPR" etc.? You do not define those terms beforehand, so you cannot expect a reader to follow your thoughts.

=> I added exact questions about  "knowledge", "confidence of CPR" etc.

-) Discussion: Try to structure your Discussion more, maybe with subheadings. Try to group ideas in "packages" and try to follow a red line throughout the text so a reader can follow your thoughts easily without becoming confused or frustrated.

=>I tried to stucture my discussion with subheadings.

-) Discussion (lines 201-202): You are right, but statements like these must be cited.

=> I added reference 22.

-) Conclusion: Try to make a statement that one can understand without having read the whole article beforehand (e.g., "of both groups" - which groups? etc.).

=> I changed to " who took either chest compression-only CPR training or conventional CPR training"

Reviewer 3 Report

Dear Authors, 

I suggested the Editors to accept the manuscript with minor revisions as follows.

Minor revision requests:

Page 1. line 38:

„A total of 29,832 OHCA victims per million peoples…” should be replaced to „ A total of 29,832 OHCA victims per million people”

Page 1. line 39:

„Since brain damage begins after four to five minutes of cardiac arrest,”. Authors should state more precisely when the irreversible brain damage would start…

Page 2 line 57

„And actually there was a report that police arrived at the scene before 119 firefight” Authors should note what 119 means, is it the number of firefirghters or tel. number of the firefighter call center.

Page 2 lines 86-89, the methods.

The two groups had the same time duration of 150 mins of theroretical and practical trainings with the only difference of ventilation training included or not. If the policemen were trained an additional ventilation practice which other part of the training was shortened? May it influence the skill retention?

Page 3 Line 111-113.

„BLS 111 sequence, consciousness check, breath check, chest compression, airway maintenance, ventilation, circulatory confirmation and brain damage.”

Why was the step „why  was the step call/shout for help” abandonned? the guidelines include this point. Are the policemen online continuously from the first call and response?

Page 3 line 120

„T tests were used to check the homogeneity between two groups,”

Authors should state that normal distribution is proven.

Page 3 line125

„SPSS for Windows ver. 20.0 was used.”

Should state the supplier company.

Page 4 Table 1.

„Frequency of Previous CPR training( n) „

The authors should explane in more detail what this exactly means, within what preceding duration from the training.

Page 4 line 139-140

„According to knowledge, mean of conventional CPR group was 5.9 (SD 1.2) and mean of chest compression-only CPR was 6.0 (SD 1.0).”

Authors may use the phrase „Concerning... knowledge points-... „ and should use mean±SD format.

Page 5, line 178

„Significant impact factors on accuracy of CPR skills were none”

I would suggest to write in this was instead: „No significant impact factors were found ...”

I would also recommend an additional paper that investigates the effect of the timing of testing method on the BLS training to be cited in the discussion:

Kovács E: The timing of testing influences skill retention after basic life support training: a prospective quasi-experimental study. BMC Med Educ. 2019 Dec 4;19(1):452. doi: 10.1186/s12909-019-1881-7.

Sincerely yours,

Author Response

Page 1. line 38:

„A total of 29,832 OHCA victims per million peoples…” should be replaced to „ A total of 29,832 OHCA victims per million people”

==>I changed "peoples" to "people".

Page 1. line 39:

„Since brain damage begins after four to five minutes of cardiac arrest,”. Authors should state more precisely when the irreversible brain damage would start…

==>I changed "brain damage begins " to "the irreversible brain damage would start".

Page 2 line 57

„And actually there was a report that police arrived at the scene before 119 firefight” Authors should note what 119 means, is it the number of firefirghters or tel. number of the firefighter call center.

==>119 is tel. number of the firefighter call center in Korea.

Page 2 lines 86-89, the methods.

The two groups had the same time duration of 150 mins of theroretical and practical trainings with the only difference of ventilation training included or not. If the policemen were trained an additional ventilation practice which other part of the training was shortened? May it influence the skill retention?

==>I meat 150 mins of theroretical and practical trainings with  chest compression and AED, additional 30 mins for ventilation training. So, I changed "150 mins ......ventilation practice" to "180 mins ....... ventilation practice (30 mins)".

Page 3 Line 111-113.

„BLS 111 sequence, consciousness check, breath check, chest compression, airway maintenance, ventilation, circulatory confirmation and brain damage.”

Why was the step „why  was the step call/shout for help” abandonned? the guidelines include this point. Are the policemen online continuously from the first call and response?

==>The question 'BLS sequence' was about the step call/shout for help.

Page 3 line 120

„T tests were used to check the homogeneity between two groups,”

Authors should state that normal distribution is proven.

==>I changed " T tests were used to check the homogeneity between two groups" to "normal distribution is proven".

Page 3 line125

„SPSS for Windows ver. 20.0 was used.”

Should state the supplier company

==>I added ( IBM Co., Armonk, NY, USA) .

Page 4 Table 1.

„Frequency of Previous CPR training( n) „

The authors should explane in more detail what this exactly means, within what preceding duration from the training

==>I changed "Frequency of Previous CPR training( n)" to "Frequency of CPR training during the past ten years(n)".

Page 4 line 139-140

„According to knowledge, mean of conventional CPR group was 5.9 (SD 1.2) and mean of chest compression-only CPR was 6.0 (SD 1.0).”

Authors may use the phrase „Concerning... knowledge points-... „ and should use mean±SD format.

==>I changed " According to knowledge, mean of conventional CPR group was 5.9 (SD 1.2) and mean of chest compression-only CPR was 6.0 (SD 1.0).” to "Concerning knowledge points, mean of conventional CPR group was 5.9±1.2 and mean of chest compression-only CPR was 6.0±1.0. "

And I changed all of expression like mean±SD.

Page 5, line 178

„Significant impact factors on accuracy of CPR skills were none”

I would suggest to write in this was instead: „No significant impact factors were found ...”

==>I changed „Significant impact factors on accuracy of CPR skills were none” to  „No significant impact factors were found”

I would also recommend an additional paper that investigates the effect of the timing of testing method on the BLS training to be cited in the discussion:

Kovács E: The timing of testing influences skill retention after basic life support training: a prospective quasi-experimental study. BMC Med Educ. 2019 Dec 4;19(1):452. doi: 10.1186/s12909-019-1881-7.

==> I added "Kovács E: The timing of testing influences skill retention after basic life support training: a prospective quasi-experimental study. BMC Med Educ. 2019 Dec 4;19(1):452. doi: 10.1186/s12909-019-1881-7." in discussion.

Reviewer 4 Report

This manuscript presents a study that assesses which is the most efficient method in training police officers in CPR for the retention of their knowledge. This is a well-designed study, although there are a few tips to keep in mind:

Introduction:

Well designed.

Why does the 2010 guidelines appear in citation 7 and not 2015?

Material and methods

Well described

I would like to assess what is the difference between the 120 minutes of practice between the two groups: What was the compression practice time in each group? What was the ventilation practice time in the conventional CPR group?

It should be mentioned that the practical tests (right after and three months after) were tests of only chest compressions, without evaluating ventilations.

Results

Well described, correct statistical analysis.

Discussion and conclusion

The authors point to the compression-only method as more efficient in retention than the conventional method. However, an assessment is being made in a supposed compression-only simulation, without assessing the ventilations. I understand that the conventional CPR training group had a shorter time to train compressions. It is normal for them to retain chest compression skills worse (if you train the compressions for 120 minutes, you will probably remember them better than if you train them for 60 minutes). If the ventilations were assessed in the simulation, it is likely that the overall quality was higher in the conventional CPR training group. That is why I believe that in the material and methods the practice times should be explained and that it should be mentioned that the assessment was only of the chest compressions in the limitations section.

I think the message from this manuscript should be that the compression only method is more efficient for retention of chest compressions. It is also discussed in the discussion of annual training recommendations. I think compression-only CPR should be promoted as the foundation of training, including conventional CPR in subsequent trainings to expand skills.

Compression-only CPR may be more efficient as a first approach to police training, later expanding with conventional CPR in future sessions.

Please keep these insights in mind to take into account those limitations and provide a clear message to readers.

Bibliography

Consider the following manuscripts as evidence in the trainings:

  • Aranda-García S. Herrera-Pedroviejo E, Abelairas-Gómez C. Basic Life-Support learning in undergraduate students of Sports Sciences: Efficacy of 150 minutes of training and retention after eight months. Int J Environ Res Public Heath. 2019;16:4771.
  • Brady WJ, Mattu A, Slovis CM. Lay responder care for an adult with out-of-hospital cardiac arrest. The New England Journal of Medicine 2019:381:2232-51.
  • Otero-Agra M, Hermo-Gonzalo MT, Santos-Folgar M, Fernández-Méndez F, Barcala-Furelos R. Assessing ventilation skills by nursing students in paediatric and adult basic life support: A crossover randomized simulation study using Bag-Valve-Mask (BMV) vs Mouth-to-Mouth Ventilation (MMV). Signa Vitae 2020; DOI:10.22514/sv.2020.16.0072
  • González-Salvado V, Fernández-Méndez F, Barcala-Furelos R, Peña-Gil C, González-Juanatey JR, Rodríguez-Núñez A. Very brief training for laypeople in hands-only cardiopulmonary resuscitation. Effect of real-time feedback. Am J of Emerg. Medicine 2016;34:993-8.
  • González-Salvado V, Rodríguez-Ruiz E, Abelairas-Gómez C, Ruando-Raviña A, Peña-Gil C, González-Juanatey JR, et al. Training adult laypeople in basic life support. A systematic review. Rev Esp Cardiol 2020;73:53-68.

Author Response

Introduction:

Well designed.

Why does the 2010 guidelines appear in citation 7 and not 2015?

==>I changed "2010 guidelines " to "2015 guidelines"

Material and methods

Well described

I would like to assess what is the difference between the 120 minutes of practice between the two groups: What was the compression practice time in each group? What was the ventilation practice time in the conventional CPR group?

==>The compression practice time in each group was 90mins. The ventilation practice time in the conventional CPR group was 30 mins. Automated external defibrillator (AED) operation practice time was 30 mins.

It should be mentioned that the practical tests (right after and three months after) were tests of only chest compressions, without evaluating ventilations.

==> I added "without evaluating ventilations".

Results

Well described, correct statistical analysis.

Discussion and conclusion

The authors point to the compression-only method as more efficient in retention than the conventional method. However, an assessment is being made in a supposed compression-only simulation, without assessing the ventilations. I understand that the conventional CPR training group had a shorter time to train compressions. It is normal for them to retain chest compression skills worse (if you train the compressions for 120 minutes, you will probably remember them better than if you train them for 60 minutes). If the ventilations were assessed in the simulation, it is likely that the overall quality was higher in the conventional CPR training group. That is why I believe that in the material and methods the practice times should be explained and that it should be mentioned that the assessment was only of the chest compressions in the limitations section.

==> I added "If the ventilations were assessed in the simulation, it is likely that the overall quality was higher in the conventional CPR training group. But the assessment of the chest compressions was done only. So there is the limitation of  overall quality.

I think the message from this manuscript should be that the compression only method is more efficient for retention of chest compressions. It is also discussed in the discussion of annual training recommendations. I think compression-only CPR should be promoted as the foundation of training, including conventional CPR in subsequent trainings to expand skills.

==>I do agree with your opinion.

Compression-only CPR may be more efficient as a first approach to police training, later expanding with conventional CPR in future sessions.

==> I agree with your opinion.

Please keep these insights in mind to take into account those limitations and provide a clear message to readers.

Bibliography

Consider the following manuscripts as evidence in the trainings:

  • Aranda-García S. Herrera-Pedroviejo E, Abelairas-Gómez C. Basic Life-Support learning in undergraduate students of Sports Sciences: Efficacy of 150 minutes of training and retention after eight months. Int J Environ Res Public Heath. 2019;16:4771.
  • Brady WJ, Mattu A, Slovis CM. Lay responder care for an adult with out-of-hospital cardiac arrest. The New England Journal of Medicine 2019:381:2232-51.
  • Otero-Agra M, Hermo-Gonzalo MT, Santos-Folgar M, Fernández-Méndez F, Barcala-Furelos R. Assessing ventilation skills by nursing students in paediatric and adult basic life support: A crossover randomized simulation study using Bag-Valve-Mask (BMV) vs Mouth-to-Mouth Ventilation (MMV). Signa Vitae 2020; DOI:10.22514/sv.2020.16.0072
  • González-Salvado V, Fernández-Méndez F, Barcala-Furelos R, Peña-Gil C, González-Juanatey JR, Rodríguez-Núñez A. Very brief training for laypeople in hands-only cardiopulmonary resuscitation. Effect of real-time feedback. Am J of Emerg. Medicine 2016;34:993-8.
  • González-Salvado V, Rodríguez-Ruiz E, Abelairas-Gómez C, Ruando-Raviña A, Peña-Gil C, González-Juanatey JR, et al. Training adult laypeople in basic life support. A systematic review. Rev Esp Cardiol 2020;73:53-68.

==> I added above mentioned manuscripts in reference.

Round 2

Reviewer 2 Report

Dear Authors,

Thank you for addressing my comments and concerns.

I feel that most of the manuscript's flaws have been amended.

However, the whole manuscript is still lacking correct English grammar and is hard to read (e.g., "in conclusionS" in the abstract, the first sentence in the Introduction, etc.). I find it hard to believe that the manuscript has been edited by an English native speaker as you state in your response to my previous comment.

The whole manuscript must either be revised by an English native speaker or a professional editing service (maybe ask the journal for help).